# Multidimensional analysis of immune responses identified biomarkers of recent *Mycobacterium tuberculosis* infection

**Tessa Lloyd**[1,2], **Pia Steigler**[1,3], **Cheleka A. M. Mpande**[1], **Virginie Rozot**[1], **Boitumelo Mosito**[1], **Constance Schreuder**[1], **Timothy D. Reid**[1], **Mark Hatherill**[1], **Thomas J. Scriba**[1], **Francesca Little**[2⊙], **Elisa Nemes**[1⊙]*, **the ACS Study Team**[¶]

**1** South African Tuberculosis Vaccine Initiative, Institute of Infectious Disease and Molecular Medicine, Division of Immunology, Department of Pathology, University of Cape Town, Cape Town, South Africa, **2** Department of Statistical Sciences, University of Cape Town, Cape Town, South Africa, **3** Wellcome Centre for Infectious Diseases Research (CIDRI) in Africa, Institute of Infectious Disease and Molecular Medicine and Division of Immunology, Department of Medicine, University of Cape Town, Cape Town, South Africa

⊙ These authors contributed equally to this work.
¶ Membership of the ACS Study Team is listed in the Acknowledgments.
* elisa.nemes@uct.ac.za

**Data Availability Statement:** Data and meta data from both groups of individuals used in this study are collated in Excel format on FigShare (https://doi.org/10.25375/uct.13693699.v3). R code used

## Abstract

The risk of tuberculosis (TB) disease is higher in individuals with recent *Mycobacterium tuberculosis* (*M.tb*) infection compared to individuals with more remote, established infection. We aimed to define blood-based biomarkers to distinguish between recent and remote infection, which would allow targeting of recently infected individuals for preventive TB treatment. We hypothesized that integration of multiple immune measurements would outperform the diagnostic performance of a single biomarker. Analysis was performed on different components of the immune system, including adaptive and innate responses to mycobacteria, measured on recently and remotely *M.tb* infected adolescents. The datasets were standardized using variance stabilizing scaling and missing values were imputed using a multiple factor analysis-based approach. For data integration, we compared the performance of a Multiple Tuning Parameter Elastic Net (MTP-EN) to a standard EN model, which was built to the individual adaptive and innate datasets. Biomarkers with non-zero coefficients from the optimal single data EN models were then isolated to build logistic regression models. A decision tree and random forest model were used for statistical confirmation. We found no difference in the predictive performances of the optimal MTP-EN model and the EN model [average area under the receiver operating curve (AUROC) = 0.93]. EN models built to the integrated dataset and the adaptive dataset yielded identically high AUROC values (average AUROC = 0.91), while the innate data EN model performed poorly (average AUROC = 0.62). Results also indicated that integration of adaptive and innate biomarkers did not outperform the adaptive biomarkers alone (Likelihood Ratio Test $\chi^2$ = 6.09, p = 0.808). From a total of 193 variables, the level of HLA-DR on ESAT6/CFP10-specific Th1 cytokine-expressing CD4 cells was the strongest biomarker for recent *M.tb* infection. The discriminatory ability of this variable was confirmed in both tree-based models.

to perform all analyses are available on FigShare as separate R scripts (https://doi.org/10.25375/uct. 14573088.v1).

**Funding:** The study was funded by the US National Institutes of Health (R21AI127121, EN). QFT testing of ACS participants was supported by Aeras and BMGF (GC 6-74: grant 37772, WH; GC12: grant 37885, WH). The South African National Research Foundation and Statistical Association of South Africa funded scholarships to TL. The funders had no role in study design, data collection and analysis, decision to publish, or preparation of the manuscript.

**Competing interests:** The authors have declared that no competing interests exist.

A single biomarker measuring *M.tb*-specific T cell activation yielded excellent diagnostic potential to distinguish between recent and remote *M.tb* infection.

## Author summary

Tuberculosis (TB) remains a leading cause of mortality in humans worldwide. TB is caused by *Mycobacterium tuberculosis* (*M.tb*) and is spread from person to person through the air. *M.tb* infection is asymptomatic, but it can progress to TB disease in some individuals, who would benefit from preventive treatment. Progression occurs more often within 1–2 years post-infection compared to remote, established infection, but recent and remote infection cannot be distinguished with the current diagnostic tools. In this study we measured many different features of immune responses in adolescents who acquired *M.tb* infection over the previous 6 months and compared them with those who were infected for at least 1.5 years. Data integration and computational modelling allowed us to identify a single feature (*M.tb*-specific T cell activation) that could accurately distinguish recent from remote *M.tb* infection. This biomarker can be measured in blood with a simple assay, and would allow targeting of preventative treatment to those at high risk of TB progression.

## Introduction

Tuberculosis (TB) is an airborne bacterial disease that is a leading cause of mortality due to an infectious agent worldwide [1]. It is estimated that about a quarter of the world's population is infected with *Mycobacterium tuberculosis* (*M.tb*), the causative agent of TB [2]. Acquisition of *M.tb* infection is generally asymptomatic and often remains undiagnosed unless serial diagnostic testing is performed. To determine *M.tb* infection status, the QuantiFERON TB (QFT) measures the level of interferon-gamma (IFN-$\gamma$), a cytokine released by T cells, upon stimulation of blood cells with two immunodominant antigens expressed by *M.tb*, early secretory antigen 6 (ESAT6) and culture filtrate protein 10 (CFP10) (here collectively termed E6C10). The highest risk of progressing to TB disease is during the first two years post-infection [3], which can be measured as recent QFT conversion by serial testing. However, serial testing for *M.tb* infection is not routinely performed in TB endemic settings. A blood-based immune signature that enables identification of recent *M.tb* infection would therefore allow targeting of preventive treatment to those at high risk of TB progression, even without serial diagnostic testing.

In order to define immunological determinants of recent *M.tb* infection, data from different arms of the immune response, namely adaptive, donor unrestricted T (DURT) and innate cell immunity were combined. Adaptive immunity consists of memory-driven antigen-specific T cell responses, such as those measured by QFT. In this study we measured functional and phenotypic features of classical *M.tb*-specific T cell responses and refer to these variables as the adaptive dataset. In contrast, innate immune cells, such as monocytes or natural killer (NK) cells, provide non-specific cellular defence mechanisms, which are more transient in nature. DURT cells display features of both adaptive and innate immune cells and bridge both arms. In this study, we included measurements of B cell, monocyte, NK and DURT cell functions in the innate dataset. We hypothesized that the integration of multiple immune measures from

the adaptive and innate immune arms, measured on the same individuals, would outperform individual data types in stratifying individuals with recent or remote *M.tb* infection.

Data integration presents several challenges, such as: i) different scales from different data types, which is typically overcome by employing data standardization or transformations methods; ii) missing values that arise due to some individuals or time points not being available in each data table, which can be meaningfully replaced using imputation methods; and iii) high dimensionality of the dataset post-integration.

Regularized regression with sparsity is a common approach for modeling high-dimensional datasets with multicollinearity. The most popular regularized regression models include Ridge Regression [4], which minimizes the residual sum of squares subject to an L2 bound; the least absolute shrinkage and selection operator (LASSO) model [5], which imposes an L1 penalty on the regression coefficients; and the Elastic Net (EN) model [6], which is a combination of the two. The latter two models are particularly advantageous as they perform both parameter estimation and feature selection simultaneously, by shrinking the effect of some coefficients to zero. If there is a group of highly correlated variables in the dataset, the LASSO model will select one of these variables at random and ignore the rest. Hence, the EN model was designed to overcome this issue. Liu et al. (2018) [7], however, observed that the standard EN approach tends to shrink all features simultaneously and does not consider differing effect sizes in predictors from different datasets. The authors hypothesized that the Multiple Tuning Parameter Elastic Net (MTP-EN) model, that allows for different degrees of shrinking for variables from different data sets, could account for the differences between each dataset and result in a model with higher predictive performance than a standard EN approach. We therefore tested whether the MTP-EN did improve the predictive performance of the integrated dataset, by directly comparing the MTP-EN and standard EN models.

Tree-based algorithms are a common collection of machine learning classification models, consisting of simple decision trees [8] or the popular random forest (RF) model [9]. A classification decision tree is a supervised model that aims to predict a target by learning decision rules from features in a dataset. Decision trees allow easy interpretation of data, clearly ranking the importance of features and relations between predictors. A downfall, however, is that they suffer from high sampling variability [10]. RF models extend decision trees by building multiple trees on bootstrapped samples of the data and merging them together for making decisions to achieve stable and accurate predictions. RF models also introduce additional randomness by considering a random subset of $m < p$ predictor variables, where $p$ is the total number of predictors in the dataset, as potential split candidates. The importance of each predictor variable can then be quantified by averaging the total amount by which the Gini Index, a measure of node homogeneity, is decreased for a split over a given predictor over all trees. A large value will be indicative of an important predictor. The misclassification error is a natural measure of performance for the RF model.

Biomarkers identified by the regression models were validated via an internal validation procedure and further confirmed using tree-based algorithms.

## Results

Based on longitudinal QFT results, we compared two groups of healthy adolescents. One group had persistent QFT+ results (four 6-monthly measurements over 18 months, n = 30) and the other experienced QFT conversion, indicative of recent *M.tb* infection (two QFT- tests 6 months apart followed by 2 QFT+ test 6 months apart, n = 29). Participants were healthy for the duration of the study, and the groups were balanced for age, sex, ethnicity, school of recruitment (indicative of socio-economic status), and TB exposure, all factors that have been

associated with a higher risk of *M.tb* infection in the larger cohort from which these participants were selected (S1 Table and [11]). No significant associations were found between these clinical parameters, which included body mass index (BMI), and group status, hence we did not need to adjust for any of these variables. Adaptive and innate immune responses were independently measured by two different flow cytometry panels on samples collected from the same participants.

An overview of the data analysis pipeline is provided in Fig 1.

## Data pre-processing

In order to successfully integrate the adaptive and innate datasets, several data pre-processing steps needed to be addressed.

Biologically meaningful variables were selected based on a combination of pre-defined criteria to identify which effector functions (cytokines) and phenotypic markers were expressed by each cell type (detailed in S1 Text). Data filtering of the adaptive immune response features using COMPASS (S1 Text Section 1.1.1 and S1 Fig) and MIMOSA (S1 Text Section 1.1.2) retained 132 out of the 259 original variables in the dataset. Further, our novel filtering method (S1 Text Section 1.1.3 and S2 Fig) identified 61 biologically meaningful innate features from the 304 variables in the raw dataset. These included 6 features of monocytes, 15 features of NK cells, 12 mucosal associated invariant T (MAIT) cell, 10 gamma-delta ($\gamma\delta$) T cell, 6 NKT cell and 12 B cell features comprised the innate dataset. Therefore, the filtered integrated dataset (innate and adaptive combined) consisted of 193 variables in total.

We standardized the raw values between the adaptive and innate dataset, which had immensely different scales, using variance stabilizing (vast) scaling and employed a multiple factor analysis (MFA)-based imputation method [12] to account for missing data points. We found MFA imputation to outperform all other imputation methods tried based on its ability to successfully replicate the distribution of the raw data (S2 Table and S3 Fig).

## Regularized regression, biomarker discovery and model validation

An MTP-EN model was built using the 193 variables of the integrated dataset in order to assess whether applying differential penalties to each dataset improved the predictive performance of the model, measured by the average area under the receiver operating characteristic curve (AUROC) on the testing data. The standard EN model and MTP-EN model yielded identically high average AUROC values (S4 Fig) and hence, in terms of predictive performance and computation time, we found no added benefit of fitting the MTP-EN model over the standard EN model for this specific dataset. The EN model was therefore used for further analyses.

EN models were subsequently built to test whether an integrated model outperforms or adds to the single dataset models. Three EN models were built to the integrated (adaptive and innate dataset combined), and to the adaptive and the innate data types separately. The final EN model built on the adaptive variables had tuning parameter values of $\alpha$ = 0.21 and $\lambda$ = 0.82 and identified three candidate biomarkers corresponding to an average AUROC value of 0.91 (S5 Fig). The biomarkers which had non-zero coefficients in the model were proportions of total Th1 cells expressing the phenotypic marker human leukocyte antigen (HLA)-DR, identified by stimulation with either E6C10 or *M.tb*-lysate, and the frequency of interleukin (IL)2 +CD107-CD154-IFN-$\gamma$- tumor necrosis factor (TNF)+ CD4+ T cells stimulated with EspC, EspF and Rv2348c (collectively termed Esp). Due to the non-parametric nature of the raw data, the two groups (recent *versus* persistent QFT+ individuals) were compared using Wilcoxon's test [13] and the raw values for these variables were found to be significantly higher

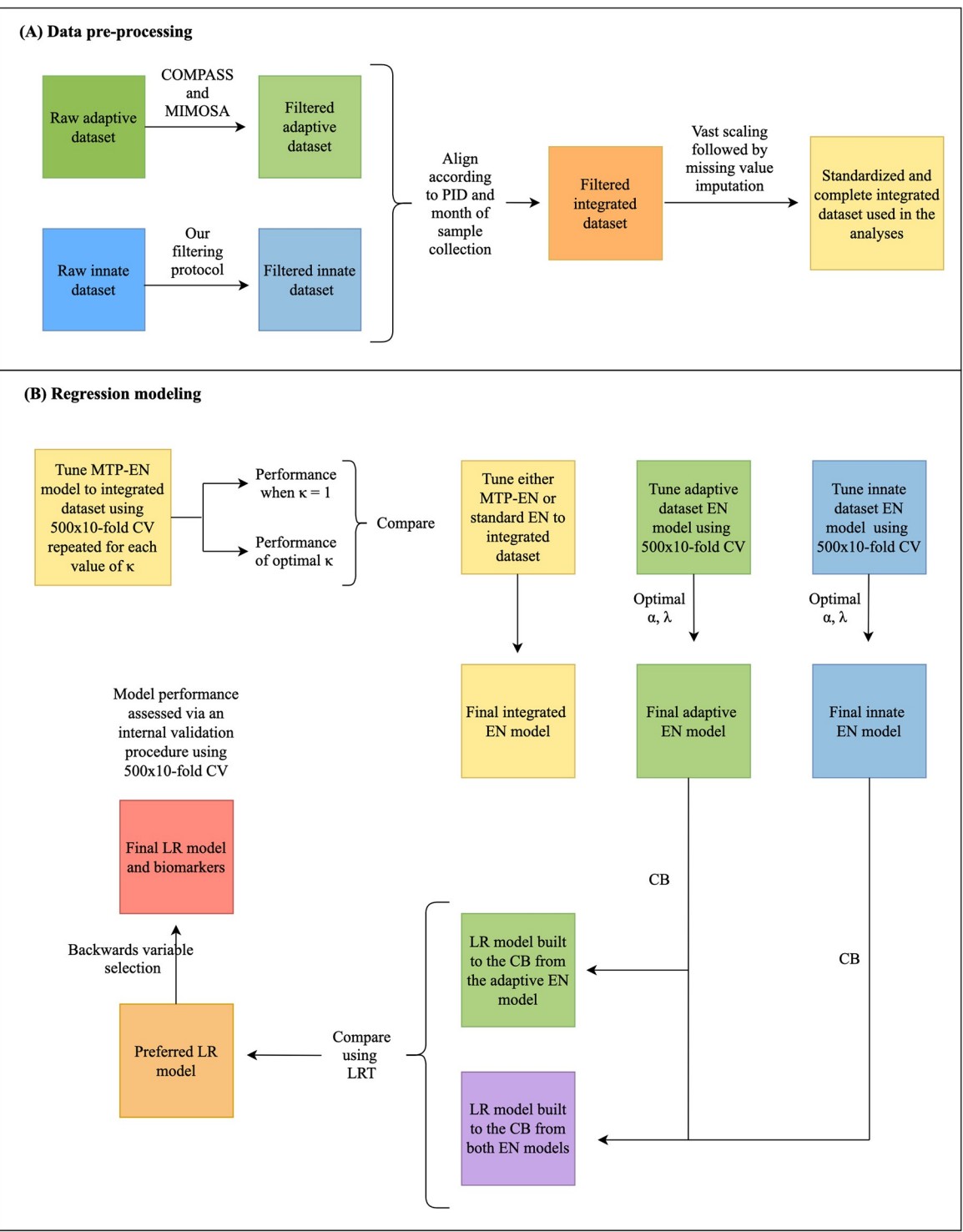

**Fig 1. Workflow showing data pre-processing steps (A) and regression modeling (B).** PID: participant ID; AUC: area under the curve; LR: logistic regression; CB: candidate biomarkers; CV: cross validation; LRT: likelihood ratio test.

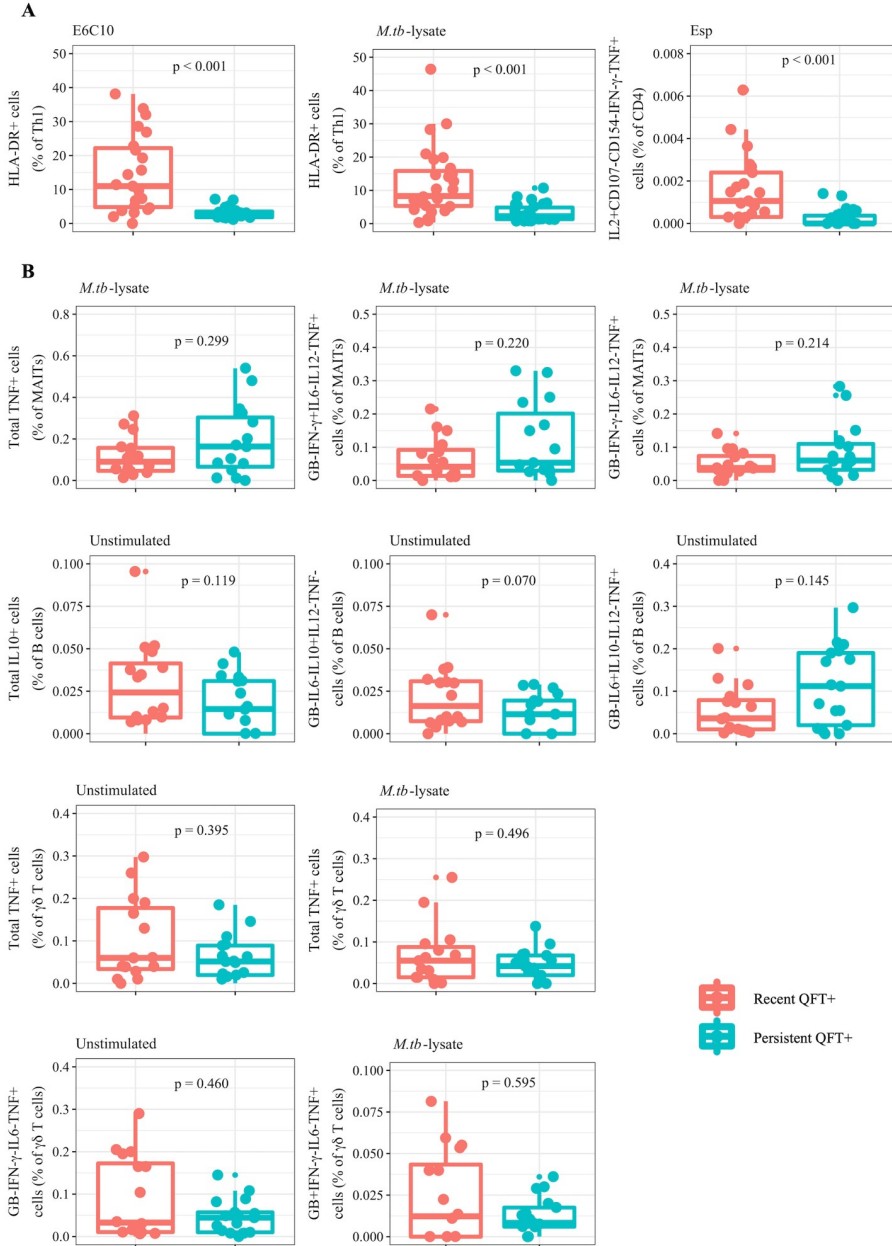

**Fig 2. Candidate biomarkers of recent *M.tb* infection identified by the adaptive and innate EN models.** Boxplots comparing the raw values of recent (red) and persistent (blue) QFT+ individuals for the three candidate biomarkers with non-zero coefficients in the adaptive EN model (A) and the 10 biomarkers with non-zero coefficients in the innate EN model (B). Wilcoxon tests were used to compare the two groups and the resulting p-values are shown.

(p < 0.001) in recent compared to persistent QFT+ individuals (Fig 2A). Hence, these variables were promising biomarkers for recent infection.

For the EN model built on the innate dataset, the optimal values for $\alpha$ and $\lambda$ were 0.18 and 0.41 respectively. This model retained 10 variables with non-zero coefficients, which yielded a poor average AUROC of 0.62 (S5 Fig). These 10 candidate biomarkers consisted of different

functional subsets of MAIT cells, B cells and $\gamma\delta$ T cells (Fig 2B). The raw values for recent and persistent QFT+ individuals for these variables were not significantly different.

The final EN model, which was built on the integrated innate and adaptive variables, yielded identical parameter values to the EN model of the adaptive data, with the same three candidate biomarkers with non-zero coefficients corresponding to an average AUROC of 0.91.

All non-zero coefficients from the integrated EN model were features from the adaptive dataset. In order to further quantitatively evaluate whether a combination of adaptive and innate features would improve the predictive performance of the model, a Likelihood Ratio Test (LRT) was used to compare a Logistic Regression (LR) model built to the 13 candidate biomarkers (three from the adaptive EN model and 10 from the innate model) to a LR model built to the three biomarkers from the adaptive EN model only. The results indicated that a combination of the non-zero coefficients from both EN models did not significantly improve model fit (LRT $\chi^2$ = 6.09, p = 0.808). The LR model built to the adaptive biomarkers was therefore the preferred model.

Backwards variable selection on this preferred adaptive LR model further identified proportions of *M.tb*-lysate-specific total Th1 cells expressing HLA-DR as a statistically redundant biomarker, since HLA-DR expression on either E6C10- or *M.tb*-lysate-specific T cells were highly correlated (S6 Fig).

The coefficients from the final LR model are shown in Table 1, model i. By exponentiating the coefficients, we can most easily interpret the coefficients in terms of the odds. Hence, holding all other variables fixed, for every one standardized unit increase in either HLA-DR or CD4+IL2+CD107-CD154-IFN-$\gamma$-TNF+ T cells in response to their specific stimuli, the odds of being a persistent QFT+ individual (the default class) decreases by 99% or 94% respectively. Accordingly, as the value of either one of these biomarkers increases, the odds that an individual was recently infected, i.e. recent QFT+, increases. The performance of this model was then assessed via an internal validation procedure and produced satisfactory results (average AUROC and Brier scores = 0.89 and 0.008 respectively).

Candidate biomarkers from the innate dataset did not improve the adaptive model fit and 2 variables from the adaptive dataset were sufficient to distinguish between the different stages of infection.

Since including the frequencies of Esp-specific IL2+CD107-CD154-IFN-$\gamma$-TNF+ CD4+ T cells as a predictor variable in the LR model statistically improved model fit (LRT $\chi^2$ = 12.76, p < 0.001) compared to E6C10-specific HLA-DR frequencies alone, we explored whether the successful discriminatory ability of this cell subset was dependent on the subset being negative for CD107, CD154 and IFN-$\gamma$. This was tested by comparing the predictive performance of an LR fitted to E6C10-specific HLA-DR and IL2+TNF+ CD4+ T cell when stimulated with Esp,

**Table 1. Model estimates and the average performance metrics after internal validation of the final LR model (i) and the LR model built to E6C10-specific total Th 1 cells expressing HLA-DR only (ii).**

|  | Coefficients | $\beta$ (95% CI) | $e^\beta$ (95% CI) | Avg. AUC | Avg. Brier |
|---|---|---|---|---|---|
| i | (Intercept) | -1.55 (-2.97; -0.13) | 0.21 (0.05; 0.88) | 0.89 | 0.008 |
|  | E6C10 HLA-DR | -4.34 (-7.20; -1.47) | 0.01 (0.00; 0.23) |  |  |
|  | Esp CD4+IL2+CD107-CD154-IFN-$\gamma$-TNF+ | -2.79 (-5.27; -0.30) | 0.06 (0.01; 0.73) |  |  |
| ii | (Intercept) | -0.91 (-1.92; 0.09) | 0.40 (0.15; 1.10) | 0.87 | 0.007 |
|  | E6C10 HLA-DR | -4.06 (-6.36; -1.76) | 0.02 (0.00; 0.17) |  |  |

regardless of CD107, CD154 and IFN-$\gamma$ expression, to model i in Table 1. We found that the average AUROC value, hence the predictive performance, of this model was lower (average AUROC = 0.79) compared to model i. In addition, this model had a higher Akaike information-tion criterion score [14] compared to model i (57 compared to 42), thus indicating a poorer fit to the data. The relative quality of the model was therefore dependent on the CD4+ T cell subset being negative for CD107, CD154 and IFN-$\gamma$.

The ability of E6C10-specific HLA-DR expression alone to distinguish between the different stages of *M.tb* infection was then assessed. The performance of this model and the model including Esp-specific IL2+CD107-CD154-IFN-$\gamma$-TNF+ CD4+ T cells were similar and equally high (average AUROC and Brier scores = 0.87 and 0.007 respectively) (Table 1, model ii).

## Tree-based methods

To verify that the final selected variables and model performance were not a result of overfitting, a simple classification tree was built to all the 193 features in the vast standardized and MFA-imputed integrated adaptive and innate dataset (Fig 3A). The tree identified two features from the set of all variables in the integrated dataset to best discriminate between recent and persistent QFT+ individuals. The best classifying feature in the dataset was the level of HLA-DR on total Th1 cells when stimulated with E6C10, followed by the frequency of Esp-specific IL2+CD107-CD154-IFN-$\gamma$-TNF+ CD4+ T cells. The split value for E6C10-specific HLA-DR was identified as -0.098 (Fig 3B). Seventeen observations had values greater than or equal to -0.098 for E6C10-specific HLA-DR expression levels and were assigned to leaf node 2, where all observations were correctly classified as recent QFT+. Otherwise, out of the seven observations in node 4, six were correctly classified as recent converters. Observations were assigned to this node if they had a value less than -0.098 for E6C10 HLA-DR but greater than -0.12 for the frequency of Esp-specific CD4+IL2+-CD107-CD154-IFN-$\gamma$-TNF+ T cells (Fig 3C). Any observations that had values less than both these split values for each of the predictors were assigned to leaf node 5, which correctly classified 29 out of the total 30 persistent QFT+ individuals, but misclassified 6 recent QFT+ individuals. These decision rules identified by the tree resulted in 12% (7 out of 59) of the observations being misclassified.

The final RF model, after hyperparameter tuning via cross validation (CV), was then built such that 500 decision trees built to 500 random bootstrapped samples of the data made up the forest, a random subset of 25 out of the 176 features were considered at each split, and each tree built was allowed no more than 10 nodes from root to terminal node to avoid overfitting. The Gini Index was then used to measure variable importance. Among the 10 top variables with the largest mean decrease were nine variables from the adaptive dataset (Fig 3D). The E6C10-specific HLA-DR variable resulted in the largest mean decrease. One single variable from the innate dataset, total TNF production in unstimulated $\gamma\delta$ T cells, was found to be the sixth most important.

After an internal model validation procedure, the average AUC for the final RF model was 0.84 and the average misclassification error was 0.12, precisely the misclassification error of the simple classification tree.

## Discussion

This study applied regularized regression modelling approaches and machine learning algorithms to identify biomarkers that could distinguish individuals with remote or recent *M.tb* infection, which is associated with higher risk of TB disease.

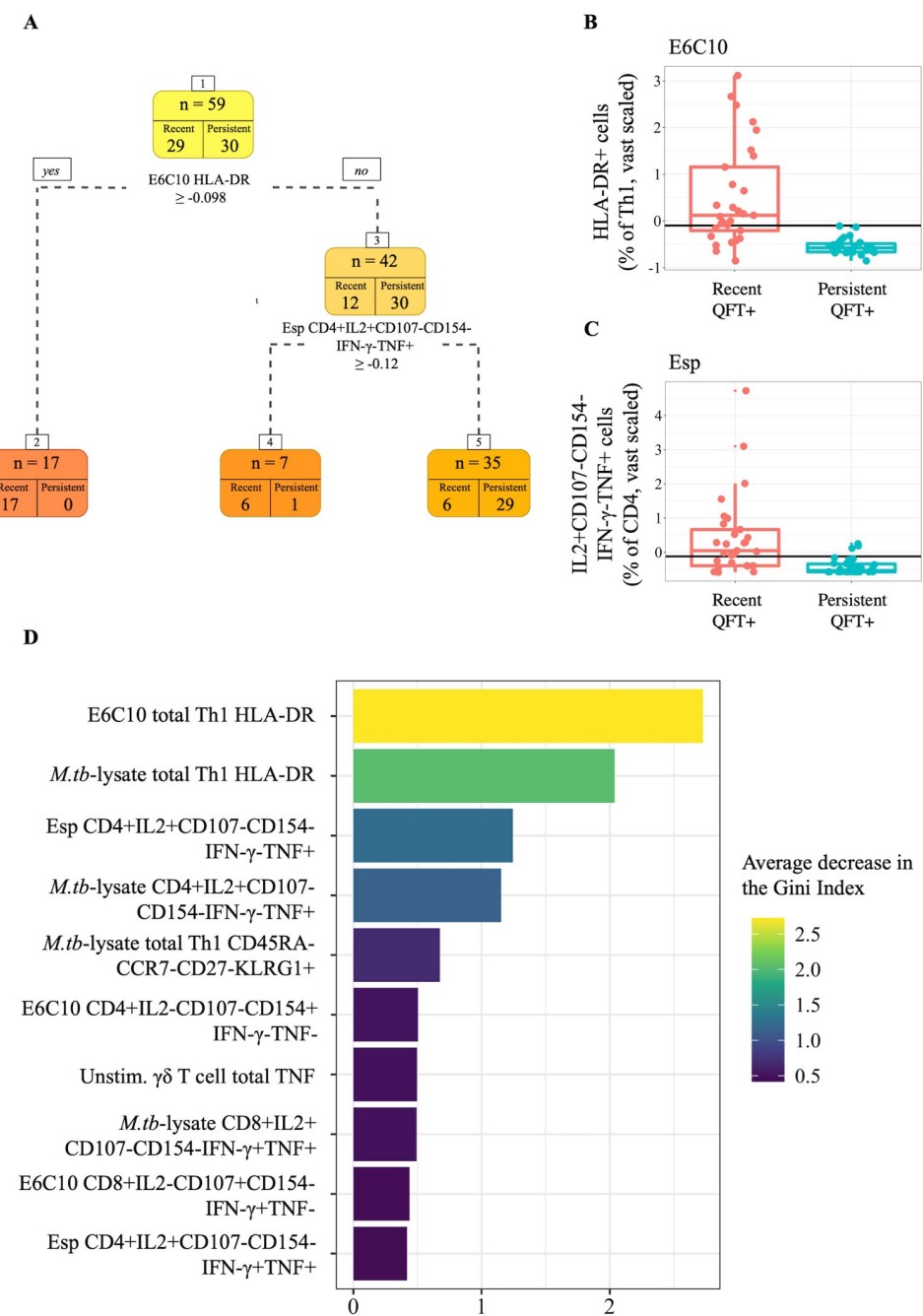

**Fig 3. Tree-based modeling results.** (A) Results from the simple classification tree built to the entire integrated dataset. Boxplots comparing vast scaled values of recent (red) and persistent (blue) QFT+ individuals were plotted for the two most stratifying features identified by the decision tree. The split values are superimposed onto the plots at (B) -0.098 for proportions of E6C10-specific Th1 cells expressing HLA-DR, and (C) -0.12 for frequencies of Esp-specific IL2+CD107-CD154-IFN-$\gamma$-TNF+ CD4+ T cells. (D) Variable importance plot of the final RF model showing the top 10 variables that resulted in the largest average decrease in the Gini Index.

Data pre-processing steps were required to successfully integrate the datasets from the innate and adaptive immune responses measured on the same individuals. To overcome the high dimensionality of the integrated dataset, filtering methods were applied to each dataset separately. COMPASS [15] and MIMOSA [16] are very sensitive algorithms that have been developed to identify biologically meaningful combinations of cytokines produced by rare antigen-specific T cells, and significant responses over background, respectively. A combination of these methods was used to pre-filter the adaptive dataset. These methods assume background immune responses detected in unstimulated samples to be extremely low. This is not the case for most innate immune cells, which spontaneously produce variable levels of cytokines, even when cultured in absence of stimulation, that can be biologically meaningful. COMPASS and MIMOSA were thus not appropriate to pre-filter innate variables and we therefore developed our own filtering method to robustly identify biologically meaningful cell subsets from innate immune cells, B cells and DURT cells.

The intrinsic biological variability between the adaptive and innate datasets was accounted for by using vast scaling to standardize the raw values to a common scale, and missing values were successfully imputed using an MFA-based imputation method. We compared several imputation methods, to account for missing values, and the MFA-based method performed the best in this dataset, characterized by non-normally distributed data with missing rows. Because our focus in this study was less on estimating model coefficients, but more on identifying predictive markers, instead of using Rubin's rule [17] to take into account imputation variability, we rather repeated the MFA imputation for each CV run. Therefore, we were confident that the results found here were not a consequence of the imputation method used. Performing such data pre-processing steps risks influencing and potentially biasing any results found in the integrated model. In this paper we hence emphasize the importance of testing different approaches and taking the time to identify the best suited method for a given dataset such that unbiased and valid results are yielded.

We first built an MTP-EN model, which applied differential penalties to the adaptive and innate datasets in the integrated model to account for potentially differing effect sizes. The results indicated that, in terms of both computing power and predictive performance, there was insufficient evidence to justify building the MTP-EN model over the standard EN model for the integrated dataset.

The EN model built to the integrated adaptive and innate dataset retained only adaptive features as non-zero coefficients, the same features that were selected by the EN model built to the adaptive dataset alone. These three candidate biomarkers were the proportions of *M.tb*-lysate or E6C10-specific Th1 cells expressing HLA-DR and the frequencies of Esp-specific IL2 +CD107-CD154-IFN-$\gamma$-TNF+ CD4+ T cells. TNF and IL2 produced by CD4+ T cells are early response cytokines that both play an important role in the context of TB [18]. HLA-DR on the other hand is a cell surface receptor reflecting T cell activation. HLA-DR expression on *M.tb*-specific T cells is an excellent biomarker to distinguish individuals with (remote) *M.tb* infection from those with active TB disease, and to monitor antibiotic treatment response [19–22]. The robustness of HLA-DR expression as a biomarker to also distinguish recent from remote asymptomatic *M.tb* infection was confirmed in response to either E6C10 or *M.tb* lysate. We propose E6C10 to be a more appropriate stimulation for use in diagnostic tools, since it only includes antigens specific for *M.tb* (the same as in interferon gamma release assays), whereas *M.tb*-lysate contains a mix of different antigens that do cross-react with other mycobacteria and is therefore less specific.

The true effect of these two identified biomarkers on the probability of an individual being remotely *M.tb* infected (persistent QFT+ individuals) was estimated through a LR model. Higher standardized frequencies of both these biomarkers were associated with a larger

probability, or odds, of an individual being recently infected (recent QFT+). This reflects the relationship that was seen in the raw data plots, where individuals recently infected with *M.tb* had significantly higher values of these features compared to remotely infected individuals. The performance of the LR model built to these two biomarkers was assessed via an internal validation procedure and, given the small sample size, was considered sufficiently high to justify further evaluation of these biomarkers.

A diagnostic test made up of a single biomarker would be simpler and more cost effective. Therefore, the ability of *M.tb*-specific T cell activation (HLA-DR expression) to successfully distinguish between the two stages of *M.tb* infection as a biomarker on its own was tested. Moving forward with HLA-DR as a single diagnostic measure was justified by substantial literature showing excellent performance of this biomarker to distinguish different stages of the TB spectrum [20–24], and the small number of markers necessary to measure this biomarker (as few as four [19]). Further, frequencies of the Esp-specific IL2 +CD107-CD154-IFN-$\gamma$-TNF+ CD4+ T cells were extremely low (values range between 0 and 0.006), which is challenging to measure in a robust and reproducible way. Lastly, because the discriminatory ability of Esp-specific IL2+CD107-CD154-IFN-$\gamma$-TNF+ CD4 + T cells was in fact dependent the subset being negative for CD107, CD154 and IFN-$\gamma$, the flow cytometry antibody panel for a diagnostic test including all these markers would be complex. Although the final LR model included frequencies of Esp-specific IL2 +CD107-CD154-IFN-$\gamma$-TNF+ CD4+ T cells as an additional biomarker of recent infection, further analyses showed that E6C10-specific HLA-DR expression alone is an equally strong single biomarker to distinguish recent from remote *M.tb* infection.

Due to the various variable selection techniques that were applied during this study to get to the final model, we run the risk of overfitting the model. The performance of the final LR model was assessed via CV and the estimated coefficient of HLA-DR in the model with multiple variables was similar to the coefficient of the model with HLA-DR alone, indicating that the relationship between HLA-DR and recent *M.tb* infection is unchanged after variable selection. However, we further applied different tree-based algorithms to ensure our approach was generalizable across statistical methods. The top performance of *M.tb*-specific T cell activation over all other immune features as a biomarker of recent infection was found in both the simple decision tree and random forest model.

Lastly, in contrast to our hypothesis, variables from the innate dataset did not improve model fit and were unable to outperform the strongest candidate biomarkers from the adaptive dataset.

To our knowledge, this study includes the most comprehensive integrated evaluation of adaptive and innate immune responses induced by recent *M.tb* infection in humans published to date. Limitations include the narrow age range of participants (13–18 years old), the unknown time of TB exposure for most individuals, the relatively wide interval between QFT testing (6 months) and the small sample size, which did not allow further stratification for risk of incident TB. Our results show that the innate immune responses were poor predictors of recent *M.tb* infection, and did not improve the performance of the integrated model. Based on the results reported here, the expression of HLA-DR on E6C10-specific T cells was the strongest candidate biomarker to distinguish between groups of participants with recent or remote *M.tb* infection. This biomarker holds the potential to identify individuals at high risk of TB progression, who would benefit from preventive TB treatment, and its performance has now been validated in a separate test cohort [24]. However, due to the small sample size in this study, further validation in a large independent cohort is required, as well as assessment of the biomarker performance in different populations, including other younger age groups, individuals living with HIV and low transmission settings.

## Materials and methods

### Ethics statement

This study was approved by the University of Cape Town Human Research Ethics Committee (protocol references: 045/2005). Written assent from participating adolescents and written consent from their parents or legal guardians was obtained prior to the study start.

### Study design and participants

An epidemiological study was carried out from July 2005 through February 2009, in which healthy, 12 to 18-year-old adolescents were recruited from local high schools in the Worcester area, Western Cape, South Africa [25, 26]. Participants who tested human immunodeficiency virus (HIV) positive, were diagnosed with TB, or had any other acute or chronic medical diseases that resulted in hospitalization during the study period, were excluded from the study. Pregnant or lactating females were also excluded. Peripheral blood mononuclear cells (PBMCs) were collected at enrolment and at 6-monthly intervals during the 2-years of follow-up (termed months 0, 6, 12 and 18) when the QFT tests were performed to determine *M.tb* infection. The QFT tests were performed and interpreted according to the manufacturer's instructions. Two groups of participants were defined based on their longitudinal QFT results and taking into account our proposed uncertainty zone to interpret quantitative values [27]: recent QFT converters (two consecutive QFT negative results, of which at least one is < 0.2 IU/mL, followed by consecutive two QFT positive results, of which at least one is > 0.7 IU/mL) and persistent QFT positives (QFT positive results ≥ 0.35 IU/mL at four consecutive visits) (S7 Fig). Raw QFT results have been described in detail elsewhere (training cohort in [24]).

Overall, recent QFT converters and persistent QFT positives were matched by age, sex, ethnicity, school (indicative of socio-economic status in our community) and known TB exposure, all factors that were associated with QFT+ in the larger cohort from which these participants were selected [11]. Since all participants were healthy for the duration of the study, no additional clinical variables could be considered to adjust the analysis.

### Definition of recent and remote *M.tb* infection

Infection with *M.tb* likely occurred between the second and third sampling occasions in the recent QFT+ individuals, which was indicated by a QFT test conversion from negative to positive. Wilcoxon's signed rank test indicated no difference over time, thus we used the median value of the two QFT positive time points for each variable in recent QFT+ individuals (n = 29 for the adaptive dataset and n = 16 for the innate dataset).

Time of *M.tb* infection was unknown in persistent QFT+ individuals (n = 30 for the adaptive dataset and n = 17 for the innate dataset). Since the Friedman's test [28] did not reveal any significant changes over time, we included median values of each variable measured at all four QFT+ time points available as representative of remote *M.tb* infection (S7 Fig).

### Immune measurements

Innate and adaptive effector responses were measured in stimulated PBMCs using flow cytometry (S1 Text Section 1.2 and [29]). Five stimulations were used to induce *M.tb* and non-specific T cell responses in cells of the adaptive arm (adaptive dataset), including *M.tb*-specific peptide pools spanning E6C10 or EspC, EspF and Rv2348c (collectively termed Esp), and *M.tb*-lysate, which is a mixture of *M.tb*-specific antigens, of which some of which are cross-reactive with other mycobacteria; Staphylococcus Enterotoxin B (SEB), as a positive control; or the cells were left unstimulated as a negative control. This dataset consisted of 259 variables

including a combination of 5 effector functions, namely interleukin-2 (IL2), CD107, CD154, IFN-$\gamma$ and tumor necrosis factor (TNF) produced by CD4+ and CD8+ T cells upon stimulation. Combinations of the phenotypic markers CD45RA, CCR7, CD27, KLRG1, HLA-DR and CXCR3 were further measured on IFN-$\gamma$, IL2 or TNF producing T cells (total Th1). Effector responses were background subtracted (subtracting the frequencies detected in corresponding unstimulated samples from frequencies in stimulated samples), while the phenotypic markers were expressed as proportions of Th1 cells. Further, phenotypes were only measured in "responding" samples (S1 Text Section 1.1.2).

In the innate dataset, effector responses were measured in unstimulated PBMC or after stimulation with *M.tb*-lysate or *Escherichia coli* (*E. coli*), which served as a positive control. The innate dataset consisted of 283 variables, including a combination of 6 functions, Granzyme B (GB), IL6, IL10, IL12, IFN-$\gamma$ and TNF produced by NK cells, B cells, monocytes, and DURT cells: mucosal associated invariant T (MAIT) cells, $\gamma\delta$ T cells and NKT cells.

## Data integration

Adaptive and innate datasets were generated independently of one another using different assays. The integration of the adaptive and innate datasets was performed by aligning each dataset according to participant ID, QFT status (positive or negative) and month of sample collection (months 0, 6, 12, and 18).

## Data pre-filtering

Due to the high dimensionality of the dataset post-integration, we opted to pre-filter the dataset to identify and exclude biologically irrelevant cell subsets. For the adaptive dataset, we employed COMPASS (Combinatorial Polyfunctionality analysis of Antigen-Specific T cell Subsets) to filter the effector functions [15] (S1 Text Section 1.1.1), while the phenotypic markers expressed on T cells were only measured in stimulated samples from responding individuals identified by MIMOSA (Mixture Models for Single Cell Assays [16]; S1 Text Section 1.1.2). Since innate immune cells and DURT cells have high background (unstimulated) values, COMPASS could not be used to filter the innate dataset. We designed a novel filtering method to identify biologically meaningful cell subsets from the innate dataset (S1 Text Section 1.1.3). All analyses were performed on the pre-filtered dataset. An outline of the data pre-processing steps is provided in Fig 1A.

## Data standardization

The intrinsic biological variability of the measurements in the separate datasets was accounted for by employing vast scaling to standardize the datasets to a common scale. Vast scaling is achieved by dividing the *Z*-score by a coefficient of variation (cv) as a scaling factor. Division by the cv, which is the sample standard deviation divided by the sample mean of each variable, gives higher importance to those variables with small relative standard deviations. Vast scaling method aims to be robust and is typically used on variables that show small fluctuations [30].

## Missing value imputation

Fewer samples were used in the innate dataset compared to the adaptive, and only n = 16 of the n = 29 recent QFT+ individuals, and n = 17 out of the n = 30 persistent QFT+ individuals, were analyzed. As not all variables were measured for each individual, row-wise missingness (missing values) arose in the final dataset as a consequence of this integration step.

Several imputation methods (S2 Table) were considered to meaningfully replace the missing values in the filtered and vast scaled dataset. The performance of each method was evaluated based on how well the imputed data could replicate the density of the vast scaled, incomplete data.

## Standard elastic net model and elastic net model with multiple tuning parameters

An outline of the workflow for the modeling portion of this study is summarized in Fig 1B.

The MTP-EN model [7] extends the standard EN model [6] by imposing separate penalties to coefficients from different data types. It achieves this via the fine tuning of the parameter $\kappa$ = $\lambda_2/\lambda_1$, where $\lambda_1$ and $\lambda_2$ are the penalties applied to the coefficients from the adaptive and innate datasets respectively, which controls the shrinkage of one data type relative to the other. The model was built to the variables in the integrated dataset using the glmnet R package [31] via the "penalty.factor" argument.

For each candidate weight parameter $\kappa \in [0.2, 1.8]$, 10-fold CV was used to tune the optimal values for $\lambda$ and $\alpha$ for this specific value of $\kappa$. The CV procedure was repeated 500 times for stable estimates and the AUROC was used as a measure of performance. The highest AUROC values after 10-fold CV were stored for each of the 500 repeats, and the performance of the MTP-EN model for each value of $\kappa$ was reported as an average of the 500 AUROC values. A parameter value of $\kappa = 1$ ($\lambda_1 = \lambda_2 = \lambda$) is equivalent to a standard EN model, and so we could directly compare the performance of the standard EN model to MTP-EN models with varying penalties applied to each dataset. The result of this experiment was used to determine whether an MTP-EN or standard EN model would be the most suitable for the integrated dataset.

Further, two EN models were built using the glmnet package to the individual adaptive and innate datasets separately. The model parameters were tuned using the same CV protocol as the MTP-EN, and the average of the selected parameters across the 500 searches were thereafter defined as the "optimal" parameter values. Relevant candidate biomarkers (CB) for classifying *M.tb* infection were identified as features with non-zero coefficients in the final model, and predictive performances in terms of AUROC values of the models were then compared.

## Logistic regression

The CBs identified from the innate and adaptive EN models were used to build LR models. One LR model was built using the biomarkers identified in the adaptive model, and another using a combination of the biomarkers identified from both the adaptive and innate data EN models. A LRT was used to assess whether adding the innate biomarkers to the LR model resulted in a statistically significant improvement in the fit of the model. Backwards variable selection was performed on the preferred LR model, as established by the LRT in the previous step, to identify the best subset of predictors and build the final LR model.

## Tree-based machine learning algorithms

**Decision trees.** A simple classification was built to all of the observations in the integrated dataset using the R package rpart [32]. The decision tree was used to visualize the relationship between the variables in the integrated dataset and assess feature importance in stratifying the recent from persistent QFT+ individuals.

**Random forest models.** We built the RF model to our data using the randomForest R library [33], and tuned the model using 500x10-fold CV. Similar to the EN models, the "optimal" hyperparameters were taken as the average across the 500 repeats and used to build the

final RF model. We then used the final RF model to identify the 10 most important features corresponding to the largest mean decrease in the Gini Index.

### Internal model validation

The predictive performance of the final LR model, and subsequently the set of biomarkers, as well as the final RF models was assessed via an internal validation procedure. We employed 10-fold CV repeated 500 times and used the AUROC and the Brier score [34] as performance metrics. Results reported are an average of the performance metrics across the 500 CV repeats.

For all instances in this study when CV was performed, the missing values in the dataset was imputed separately for the training and testing sets using MFA imputation. Therefore, the dataset was imputed several times to ensure that any results found were not just a consequence of the imputation method.

## Supporting information

**S1 Table. The clinical parameters of the two groups of individuals in this study.**
(PDF)

**S2 Table. The various imputation methods that were tested.**
(PDF)

**S1 Text. Supplementary methods.** Supporting information on PBMC isolation, stimulation and staining, and on the data pre-filtering methods.
(PDF)

**S1 Fig. Filtering the adaptive dataset.** The number of observations for CD4+ T cell counts stimulated with E6C10 in the recent QFT+ individuals that had posterior probabilities (calculated by COMPASS) greater than 0.1 for each binary combination, stratified according to month. A subset was classified as biologically meaningful if the number of observations with posterior probability values greater than 0.1, at one of either month 0, 6, 12 or 18, was greater than 10 (one third of the number of participants in one cohort).
(TIF)

**S2 Fig. Flow chart for the innate data filtering.**
(TIF)

**S3 Fig. Data imputation methods.** The efficacy of each imputation method to capture the distribution of the raw frequencies of total IFN-$\gamma$ production in NKT cells stimulated with M.tb-lysate is shown as an example. The red lines are the raw data in each plot and the blue lines are (A) MFA, (B) column median, (C) $k$-nearest neighbours and (D) missForest imputed values.
(TIF)

**S4 Fig. Performance of the MTP-EN model.** The average of 500 AUROC values is plotted as a function of $\kappa$, the ratio of the penalty parameter for the innate dataset relative to that for the adaptive dataset. When $\kappa < 1$ ($\lambda_2 < \lambda_1$) a smaller penalty is applied to the innate dataset, and when $\kappa > 1$ ($\lambda_2 > \lambda_1$) a larger penalty is applied to the innate dataset. A red dashed line is plotted at $\kappa = 1$ ($\lambda_2 = \lambda_1$), which is equivalent to a standard EN model, and a blue line at the "optimal" $\kappa = 1.7$, corresponding to the highest mean AUROC.
(TIF)

**S5 Fig. Coefficient paths for the final adaptive and innate EN model as a function of log ($\lambda$).** Each line in the plots represents the coefficients of one variable for different values of $\lambda$, the overall shrinkage parameter in the EN model, from the respective datasets. An increasing

value of $\lambda$ leads to the shrinkage of more regression coefficients and the number of non-zero coefficients for each $\lambda$ value are shown at the top of the figure. In the adaptive EN model (A) $\alpha$ was set to 0.21, where $\alpha \leq 1$ is the weight given to the L1 penalty and (1- $\alpha$) the weight to the L2 penalty. A dotted line is plotted at $\log(\lambda)$ = -0.2 ($\lambda$ = 0.82), the optimal parameter values from the final adaptive EN model. At this point the number of non-zero coefficients are three and correspond to E6C10-specific or *M.tb*-lysate-specific HLA-DR expression on total Th1 cells and Esp-specific CD4+IL2+CD107-CD154-IFN-$\gamma$-TNF+ T cells. For the final innate EN (B) $\alpha$ was set to 0.21 and a dotted line is plotted at $\log(\lambda)$ = -0.89 ($\lambda$ = 0.41) corresponding to 11 non-zero coefficients.
(TIF)

**S6 Fig. Correlation between *M.tb*-lysate (x-axis) and E6C10 (y-axis) stimulation on total Th1 cells expressing HLA-DR.** Spearman's non-parametric correlation coefficient and its associated p-value are superimposed onto the plot.
(TIF)

**S7 Fig. Cohort definition.**
(TIF)

# Acknowledgments

We are thankful to the ACS study participants and their families; and the SATVI clinical and laboratory teams.

ACS Study Team: Hassan Mahomed, Willem A. Hanekom, Fazlin Kafaar, Leslie Workman, Humphrey Mulenga, Ashley Veldsman, Rodney Ehrlich, Mzwandile Erasmus, Deborah Abrahams, Anthony Hawkridge, E. Jane Hughes, Sizulu Moyo, Sebastian Gelderbloem, Michele Tameris, Hennie Geldenhuys, Gregory Hussey.

# Author Contributions

**Conceptualization:** Virginie Rozot, Mark Hatherill, Thomas J. Scriba, Elisa Nemes.

**Data curation:** Tessa Lloyd, Pia Steigler, Cheleka A. M. Mpande.

**Formal analysis:** Tessa Lloyd, Francesca Little.

**Funding acquisition:** Mark Hatherill, Thomas J. Scriba, Elisa Nemes.

**Investigation:** Pia Steigler, Cheleka A. M. Mpande, Boitumelo Mosito, Constance Schreuder, Timothy D. Reid.

**Methodology:** Francesca Little.

**Project administration:** Elisa Nemes.

**Resources:** Elisa Nemes.

**Supervision:** Pia Steigler, Virginie Rozot, Thomas J. Scriba, Francesca Little, Elisa Nemes.

**Visualization:** Tessa Lloyd.

**Writing – original draft:** Tessa Lloyd, Pia Steigler, Francesca Little, Elisa Nemes.

**Writing – review & editing:** Tessa Lloyd, Pia Steigler, Thomas J. Scriba, Francesca Little, Elisa Nemes.

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
