## [Decision Letter · Decision Letter 0]

23 Apr 2021

Dear Dr Nemes,

Thank you very much for submitting your manuscript "Multidimensional analysis of immune response identified  biomarkers of recent Mycobacterium tuberculosis infection" for consideration at PLOS Computational Biology.

As with all papers reviewed by the journal, your manuscript was reviewed by members of the editorial board and by several independent reviewers. In light of the reviews (below this email), we would like to invite the resubmission of a significantly-revised version that takes into account the reviewers' comments.

The results are interesting but a number of clarifications is needed. Please take into account all the reviewers comments. The cohort/patient population should be better described. It would be also beneficial for the paper to validate the results on an independent cohort. It also should be discussed which clinical parameters are used to correct the statistical analysis, and if no adjustments are done, explain why. It is also important to verify whether the model is not overfitted.

We cannot make any decision about publication until we have seen the revised manuscript and your response to the reviewers' comments. Your revised manuscript is also likely to be sent to reviewers for further evaluation.

Sincerely,

Nataliya Sokolovska

Guest Editor

PLOS Computational Biology

Thomas Leitner

Deputy Editor

PLOS Computational Biology

The results are interesting but a number of clarifications is needed. Please take into account all the reviewers comments. The cohort/patient population should be better described. It would be also beneficial for the paper to validate the results on an independent cohort. It also should be discussed which clinical parameters are used to correct the statistical analysis, and if no adjustments are done, explain why. It is also important to verify whether the model is not overfitted.

Reviewer's Responses to Questions

**Comments to the Authors:**

Reviewer #1: Review: Multidimensional analysis of immune response identifies biomarkers of recent Mycobacterium tuberculosis infection

This paper attempts to identify immune factors associated with recent TB infection, as opposed to long-term latent TB. The authors investigate TB-specific and non-specific responses in both innate and adaptive cell types in a cohort of longitudinally monitored South African adolescents. In particular, they point to an increase in HLA-DR+ T cells specific for ESAT-6 and CFP-10 and IL2+CD107-CD154-IFNg+TNF+ specific for EspC, EspF and Rv2348c as key markers of recent TB infection. These findings are both novel and important. However, the paper has a number of flaws, including a lack of clarity in the Results on the study cohort(s) and immune filtering criteria. The models described here were evaluated by cross validation, without an independent validation cohort. Without independent validation it is difficult to know how replicable these results would be in another cohort.

Major comments

Results

Line 124: Overall, the results section is lacking in important details. Line 124 states that “in order to successfully integrate the two datasets”. This is unclear. What are the datasets being referred to here? Innate and adaptive? One study dataset is described in the Methods, but a basic overview should be in the results. These include, numbers of participants, what samples were taken and what type of assays were run. The criteria for filtering the datasets (see also line 424 in methods – more detail needed) and selecting the imputation method should also be stated, these are key points for understanding the study and this information should not be relegated to supplemental methods.

Line 133: The MTP-EN model. The authors perform MTP-EN as well as standard elastic-net regression. MTP-EN provided no benefit over the elastic net, so I do not think this merits its own section. This could probably be relegated to a sentence in the subsequent section saying that MTP-EN does not provide any benefit.

Line 154: The final EN models … identified three candidate biomarkers. What does ‘identified’ mean here precisely? Are these three markers the only non-zero coefficients in the model? See also Line 173, with 10 biomarkers

Line 159: The authors note that HLA-DR+ Th1 cells respond to either E6C10 or total Mtb lysate. What about the IL2+CD107-CD154-IFNg-TNF+CD4+ T cells? Do these also respond to TB lysate? Overall, do cell types that respond to specific TB antigens also respond to total lysate? This is an important control.

Line 230: Statistical Validation. The authors apply an alternate classification tree-based approach to the data and term this statistical validation. I do not understand how this validates their results. A true validation would require prediction on a separate set of data held out of the training process, and training another model on the same data to get a similar result is of little value. See also Line 348 in the discussion.

Methods

Line 377: “Two cohorts of participants” Cohort normally refers to the entire group of particpants in a study. The authors seem to mean “classes” rather than cohorts.

Line 385: The definition of recent QFT converters was limited to those who showed two negative QFTs followed by two positive QFTs. It is also possible that one negative followed by three consecutive positives or three negative followed by one QFT positive are recent converters. What would happen if these individuals were included in the study?

Line 418: “Integration was performed by aligning each dataset” But there is only one study cohort mentioned – what are the datasets that have to be aligned?

Reviewer #2: Review of Lloyd et al.

The manuscript provides a very detailed account of TB data analysis based on immuno markers of various kind. Their main results is the identification using ML techniques of one biomarker to discriminate between recent and persistent interferon measures.

Main comment: While the results are interesting and should be published, I find the manuscript very hard to follow overcrowded with details and key features missing. Below in random order the various comments/criticisms I have made for the manuscript:

The cohort is not well described and all the analyses are based on it. Even if great care has been taken to select the participant the study lacks details and statistical analyses to assure that other confounding factors does not bias the dataset.

Related to previous comment: since all statistical analyses are performed based on the immune data only, no clinical parameters were introduced to correct the statistical analyses. This should be addressed either before hand or during the results comparisons.

It is not clear why the data integration is a challenge here since we seem to look at immuno data coming from the same experiments. There does not seem to have some batch effects nor clinical data. So i don’t understand.

There is no reason to justify Wilcoxon test as it merely stacks the order. Non parametric statistical analysis should be avoided especially since the authors explicitly compare the means.

I’m not well aware of the debate between various elastic net strategies and why it should be the main focus of the publication (and take 80% of the abstract). I suggest to streamline the manuscript to make the biological results more understandable and more the center of the manuscript. Especially as far as I understood the main result is the figure 3.A. and the fact that additional innate marker yield no further information. Or did I miss something?

Reviewer #3: The authors investigated an important topic in the field of Infectious diseases. As they have pointed out the risk of tuberculosis disease is higher in individuals with recent Mycobacterium tuberculosis infection compared to individuals with more remote, established infection. Therefore, the development of tools to distinguish this patient’s population which is at higher risk for the development of TB disease could be key for disease prevention strategies.

Based on the investigators results, the determination of the level of HLA-DR on ESAT6/CFP10-specific Th1 cytokine-expressing CD4 cells has the potential to serve as a strong biomarker for recent Mycobacterium tuberculosis infection. These findings are interesting, although they require validation in larger cohorts before they can be used for clinical management.

The tree-based models showed that for E6C10-specific HLA-DR expression levels 17 out of 29 recent QFT converters could be correctly classified with values greater than or equal to -0.098. In this model, the use of this biomarker could be considered highly specific. However, the specificity for the other nodes appeared to be lower. As the decision rules identified by the tree resulted in 12% of the observations being misclassified, the strategy proposed in this manuscript could result in false classification of recent QFT converters. Nonetheless, if a significant amount of recent QFT converters can be identified (with an average AUC for the final RF model of 0.84), this strategy has the potential to be used clinically in recently exposed patients. However, if the results of this biomarker strategy suggest that a patient is not a recent QFT converter, these results should be interpreted with caution due to an average misclassification error of 0.12.

As the authors have shown in the discussion, their group and other, have shown that HLA-DR expression on Mycobacterium tuberculosis-specific T cells has been shown to have good potential as a biomarker to distinguish individuals with LTBI from those with active TB disease, and to monitor antibiotic treatment response. These results recapitulate the fact that HLA-DR expression should be studied in broader clinical samples to determine its clinical sensitivity and specificity for the detection of populations at higher risk of developing TB disease. The relevance of this biomarker also has biological sense, as HLA-DR is a cell surface receptor reflecting T cell activation.

I agree with the authors that a diagnostic test made up of a single biomarker would be simpler and more cost effective, and therefore based on the data presented in this manuscript the use of HLA-DR expression alone for the prediction of QFT recent conversion would be preferred in clinical labs. The use of complex antibody panels may be complicated and not feasible in different settings.

Comments:

-This is a paper of Tuberculosis diagnostics. However, the demographics and clinical characteristics of the subjects are not clearly described in the paper. The authors must describe extensively in a paragraph the population, and if possible, describe the clinical characteristics in a table. The manuscript would not be acceptable for publication if this point is not addressed.

-Would it be possible to assess if a combination of Innate and adaptive effector responses measured in stimulated PBMCs using flow cytometry, and clinical variables may help improving the discriminatory value of the tree-based models. I don’t necessarily expect this to be the case, and larger cohorts may be needed to answer this question, but some individual-specific variables such as time since tuberculosis exposure, nutritional status, and receiving preventive therapy may influence the QFT conversion patterns. The authors should at least describe these variables in this patient population. It would be important to describe if these variables are different in the patients that are not correctly classified.

Questions for the authors:

-Why did the authors limit the study population to adolescents 12-to-18 year of age?

-How did the authors select the intervals for QFT testing during the 2-years of follow-up (termed months 0, 6, 12 and 18) to determine Mycobacterium tuberculosis infection? What was the rationale to select those intervals? Was it randomly selected, or there was some rationale behind? Also, what was the rationale to only follow up patients for 2 years? There is recent literature suggesting that the risk of development of TB in pediatric populations (including adolescents) is highest within 90 days of enrolment (and presumably of exposure) (2·9% [95% CI 1·7–4·9]) (Martinez, Lancet 2020). Therefore, the authors may have missed an important timeframe for biomarker discovery as testing within the first 90 days was not performed in this study. This is a very important limitation and has to be discussed in the paper. If the authors design future studies on this matter they should take this point into account.

-Can the authors describe briefly their protocol for cryopreservation of PBMCs? How long (average) were the cells stored? Was this protocol followed for all the patients? Are there any differences in storage times within the patients? This should be included in the supplemental material.

Strengths:

-The authors have defined a population longitudinally based on the QFT conversion patterns. The authors defined recent QFT converters, as individuals with two consecutive QFT negative results followed by consecutive two QFT positive results, which makes this a very well-defined population. Moreover, the need for two consecutive QFT negative and then positive results, decreases the risks of false positive and false negative results. This is a unique population which is ideal for this study.

-The authors have developed their own filtering method to robustly identify biologically meaningful cell subsets as COMPASS and MIMOSA may not be appropriate to pre-filter innate variables.

Limitations:

-The authors only studied Mycobacterium tuberculosis infected adolescents. As it has been extensively described in the literature, the natural history of disease in tuberculosis is impacted by the age of the individuals, and well-defined phenotypes occur at different ages. Adolescents usually present with adult-type disease (Martinez, Lancet 2020; Perez-Velez NEJM 2012). Therefore, the findings reported in this study may not be applicable to other patient populations of different ages. This should be described extensively in the discussion. Presumably the prediction value of the biomarkers described in this study may not work for younger patients, or even for adult populations.

**Have all data underlying the figures and results presented in the manuscript been provided?**

Reviewer #1: Yes

PLOS authors have the option to publish the peer review history of their article (what does this mean?). If published, this will include your full peer review and any attached files.

Reviewer #1: No

Reviewer #2: No

Reviewer #3: **Yes: **Julian A Villalba

**Have the authors made all data and (if applicable) computational code underlying the findings in their manuscript fully available?**

Reviewer #2: **No: **i could not find but maybe there is

Reviewer #3: Yes
---

## [Decision Letter · Decision Letter 1]

18 Jun 2021

Dear Dr Nemes,

We are pleased to inform you that your manuscript 'Multidimensional analysis of immune response identified  biomarkers of recent Mycobacterium tuberculosis infection' has been provisionally accepted for publication in PLOS Computational Biology.

Best regards,

Nataliya Sokolovska

Guest Editor

PLOS Computational Biology

Thomas Leitner

Deputy Editor

PLOS Computational Biology

The authors should prepare the final version taking into consideration the remarks of the reviewers.

Reviewer's Responses to Questions

**Comments to the Authors:**

Reviewer #1: The authors have satisfactorily addressed my comments.

Reviewer #2: The revisions are fine

Reviewer #3: I thank the authors for answering my questions. The manuscript has improved considerably, and I congratulate the authors for their study as it is an important contribution for the field. I have few minor comments:

-Thank you for clarifying about the cohort. The comprehensive description of the cohort is very helpful. It is also helpful to know that the results have been validated and published [PMID: 33406011]. Thanks also for adding the supplementary Table 2 showing demographic information. It would be helpful to add p-values stating that there are non significant differences between Persistent QFT+ and Recent QFT+.

-Thanks also for providing the reference [11] that studied all factors that were associated with a higher risk of M.tb infection in the larger cohort from which these participants were selected. From a looking at that publication, predictive factors found in multivariable analysis to be associated with positive QFT [table 2 ref 11] also included: Chronic allergy-related conditions, Income ⩽ZAR4000/month and paternal and maternal highest education level. The authors do not need to include this data, unless available. This is a minor thing, but if they have the data I encourage them to add it to supplemental table 2.

-Thanks also for adding a brief description of the study groups.

-I agree with the authors that it would be therefore very difficult (i.e. a massive sample size would be required) to detect QFT conversion in adults, since most individuals are already QFT+, but when I asked the question I was actually referring more to younger populations that are non-exposed and can represent an important population to study for Mtb infection. I think the authors should state that it is a limitation of the study that other younger age groups were not included.

-I know understand the rationale behind the interval selection for QFT testing during the 2-years of follow-up (termed months 0, 6, 12 and 18); and it is totally fine that then intervals were selected based on feasibiity. I understand that the paper did not aim nor was designed to define biomarkers for incident TB. However, in the abstract the authors mention "We aimed to define blood-based biomarkers to distinguish between recent and remote infection, which would allow targeting of recently infected individuals for preventive TB treatment". Therefore, I still think that the authors should state in their discussion that testing within early exposure was not done (as the intervals are 6-month long) and therefore these results cannot be used to predict incident TB, but only to distinguish between groups of patients with recent or remote infection.

-Thanks for adding the protocol for cryopreservation of PBMCs in the supplemental material.

**Have the authors made all data and (if applicable) computational code underlying the findings in their manuscript fully available?**

Reviewer #1: Yes

Reviewer #2: Yes

Reviewer #3: Yes

PLOS authors have the option to publish the peer review history of their article (what does this mean?). If published, this will include your full peer review and any attached files.

Reviewer #1: No

Reviewer #2: **Yes: **Hedi A. Soula

Reviewer #3: **Yes: **Julian A. Villalba

---

## [Editor Report · Acceptance letter]

23 Jul 2021

PCOMPBIOL-D-21-00187R1 

Multidimensional analysis of immune responses identified biomarkers of recent *Mycobacterium tuberculosis* infection

Dear Dr Nemes,

I am pleased to inform you that your manuscript has been formally accepted for publication in PLOS Computational Biology. Your manuscript is now with our production department and you will be notified of the publication date in due course.

With kind regards,

Zita Barta
